# Microstructure and Mechanical Properties of β-Titanium Ti-15Mo Alloy Produced by Combined Processing including ECAP-Conform and Drawing

**DOI:** 10.3390/ma15238666

**Published:** 2022-12-05

**Authors:** Svetlana A. Gatina, Veronika V. Polyakova, Alexander V. Polyakov, Irina P. Semenova

**Affiliations:** 1Laboratory of Multifunctional Materials, Ufa University of Science and Technology, 450076 Ufa, Russia; 2Department of Mechanical Engineering Innovative Technologies, Perm National Research Polytechnic University, 614990 Perm, Russia

**Keywords:** metastable β-titanium alloys, ultrafine-grained structure, mechanical properties, fatigue properties, phase transformations

## Abstract

At present, researchers pay great attention to the development of metastable β-titanium alloys. A task of current importance is the enhancement of their strength and fatigue properties. An efficient method for increasing the strength of such alloys could be severe plastic deformation. The object of this study was a medical metastable β-titanium alloy Ti-15Mo (ASTM F2066). The alloy in the (α + β) state was for the first time deformed by combined processing, including equal channel angular pressing-conform and drawing. Such processing enabled the production of long-length rods with a length of 1500 mm. The aim of the work was to study the effect of the combined processing on the alloy’s microstructure and mechanical properties. An ultrafine-grained structure with an average size of structural elements less than 100 nm was obtained. At the same time, high strength and ductility (σ_uts_ = 1590 MPa, δ = 10%) were achieved, which led to a record increase in the endurance limit (σ_−1_ = 710 MPa) under tension-compression terms.

## 1. Introduction

Metastable β-titanium alloys are a promising material in the aerospace, automotive, and biomedical industries due to their high specific strength, excellent corrosion resistance, good deformability, wear resistance, and low modulus of elasticity [1,2,3]. However, one of the important requirements for materials used in these areas is high mechanical and fatigue properties.

An increase in strength in β-titanium alloys, as well as in (α + β) alloys, is achieved by combining various mechanical and thermal treatments [4]. Several cycles of deformation in the β and (α + β) regions and recrystallization annealings make it possible to achieve the formation of a structure with small-sized β grains [5]. In addition, deformation significantly increases the response of alloys to aging, accelerates the kinetics of precipitation of the secondary α-phase, and leads to its more uniform distribution due to an increase in the number of nucleation sites, which are dislocations, grain/subgrain boundaries, and other defects [6,7,8,9]. In addition, as shown in [10,11], high applied pressures additionally initiate the α→ω and ω→β phase transformations. In [3], a review was undertaken of the fabrication, processing, and surface modification techniques for the improvement of the mechanical and functional properties of biomaterials, including β-titanium alloys. Thus, the structural-phase factors affecting the mechanical properties of metastable β-alloys include: the size of β-grains, the alloying element amount, the surface characteristics, the length of interphase and intergrain boundaries, the density of defects in the crystal lattice, morphology, distribution, volume fraction, and coherence of particles of the second phases [3,10,11,12,13,14,15].

As is known, severe plastic deformation (SPD) is an effective method for introducing crystal lattice high-density defects, such as high-angle and low-angle boundaries or dislocations, into a material [16,17]. This method makes it possible to achieve high strength combined with increased ductility and, consequently, high fatigue properties of metals and alloys due to the formation of ultrafine-grained (UFG) structures in them. Thus, an increase in the endurance limit compared to the coarse-grained state is characteristic of many light UFG metals and alloys produced by SPD [18,19]. For example, in [20,21,22,23,24,25], an increase in the endurance limit and ultimate strength of such biomedical materials as Ti Grade 4 and Ti-6Al-4V ELI was demonstrated due to the formation of a UFG structure in them.

In this work, the research material was a metastable titanium alloy Ti-15Mo (ASTM F2066), used in traumatology, orthopedics, dentistry, and cardiovascular surgery [26,27]. The alloy has the ability to change mechanical properties in a wide range of values depending on the structural-phase characteristics, which are controlled by the regimes of thermomechanical treatment (temperature, speed and time of heating/cooling, temperature, speed and degree of deformation) [28,29]. In this regard, the Ti-15Mo alloy is used in two states, depending on the preference for its physical and mechanical properties [26]. The alloy is used in the single-phase β-state, if low modulus, high ductility and good formability are needed. The alloy is used in the two-phase state, if a compromise between ductility and strength is necessary to achieve high values of fatigue properties. In previously published works, the influence of high-pressure torsion (HPT) and equal-channel angular pressing (ECAP) on the evolution of the microstructure, the kinetics of phase transformations, and the mechanical properties of titanium alloys in the β state were studied [7,8,9,10,30,31,32,33]. It was shown that the formation of a UFG structure leads to a more uniform diffusion decomposition of the solid solution and, as a consequence, to a uniform distribution of particles of the α_s_ phase of equiaxed morphology. Moreover, the authors of the article showed the possibility of achieving a high endurance limit while maintaining a low elastic modulus (σ_−1_ = 640 MPa and E = 95 GPa) relative to (α + β)-titanium alloys, due to the formation of a UFG structure by the ECAP method [33].

However, severe plastic deformation of metastable β-titanium alloys treated for a solid β-solution is accompanied by phase transformations, in particular, the stress-induced ω-phase precipitates, which embrittle the alloy [34]. In this regard, the material of this study was the Ti-15Mo alloy in a more stable two-phase (α + β) state.

In addition, most of the previous studies of the influence of a UFG structure on the mechanical properties of β-titanium alloys were carried out on the material produced by the methods of HPT, ECAP, and rotary swaging, which make it possible to obtain small laboratory samples. This article demonstrates the possibility of forming a UFG structure and a unique combination of strength and ductility in bulk rods up to 1500 mm long suitable for practical use. For this, for the first time, combined thermomechanical processing was applied to the Ti-15Mo alloy, including equal-channel angular pressing according to the “Conform” scheme (ECAP-C) and drawing [35].

Thus, the purpose of this article was to study the effect of the combined thermomechanical treatment, including ECAP-C and drawing, of the Ti-15Mo alloy in the two-phase (α + β) state on its microstructure and mechanical and fatigue properties.

## 2. Materials and Methods

The object of the study was rods made of Ti-15Mo alloy in the two-phase (α + β) state, produced by rolling in the two-phase region with subsequent slow cooling from a temperature of 788–982 °C (manufactured by Dynamet Carpenter, Richmond, VA, USA). The chemical composition of the alloy is presented in Table 1 [26].

In order to ensure the deformability of the ingots during ECAP-C, preliminary annealing was carried out at a temperature of 600 °C for 1 h, followed by water quenching (WQ).

To produce long rod semi-finished products from Ti-15Mo alloy with the UFG structure, the method of continuous ECAP according to the “Conform” scheme was used. The deformation of rods with the two-phase (α + β) structure with a diameter of 12 mm and a length of 500 mm was carried out on an ECAP-C-600 die-set (Russia), with square channels with an aspect ratio of 11 × 11 mm. The channel intersection angle was 120°. Four passes were performed: two at a temperature of 350 °C, and two at a temperature of 300 °C. The deformation was carried out according to the route B_c_ (after each cycle, the ingot was rotated in the same direction by 90° around its longitudinal axis), and the accumulated degree of deformation after four passes was e ≈ 2.8. A schematic diagram of ECAP-C is shown in Figure 1a. In the center of the fixed part of the installation, there is an engine, onto the torsion shaft of which is placed a friction wheel with a groove around the entire circumference. Due to the friction forces arising at the point of three-sided contact of the deformable rod with the pulley, the ingot is set in motion and brought to the place where it meets the element of the fixed part of the die-set. Being under the action of friction forces, the ingot, moving into a channel approaching the rotating wheel at an angle ψ, undergoes shear deformation, as in traditional ECAP [35].

For additional hardening and shaping of a rod with a round section, drawing of ingots was carried out (Figure 1b). In order to prevent the processes of recovery and recrystallization during deformation and achieve the planned level of strength, drawing was carried out at the same temperature as ECAP-Conform, i.e., at 300 °C. The drawing speed was 0.95 mm∙s^−1^. The compression ratio was 60%. As a result of ECAP-C and drawing, long-length semi-finished products with a diameter of 6 mm were produced in the form of round rods with a length of 1500 mm. The total imposed strain value on the final rod was e ≈ 4.3. Further, in order to increase the ductility of the alloy and reduce internal stresses, annealing was carried out at a temperature of 200 °C for 1 h, and cooling was carried out in air. Annealing was performed in Nabertherm N 17/HR (Nabertherm GmbH, Lilienthal, Germany) furnaces with a built-in thermocouple and an automatic control unit. The temperature gradient in the chamber did not exceed 2 °C.

Microstructural studies of Ti-15Mo alloy samples were performed using scanning electron microscopy (SEM), JSM-6390 (JEOL, Tokyo, Japan) in secondary and reflected electron modes, as well as transmission electron microscopy (TEM) JEM-2100 (JEOL, Tokyo, Japan) at an accelerating voltage of 200 kV. We used standard techniques for obtaining bright-field and dark-field images, as well as microdiffraction patterns. Thin foils were produced by jet polishing on a Tenupol-5 machine (Struers, Ballerupcity, Denmark). In this case, an electrolyte for titanium was used (5% perchloric acid, 35% butanol, and 60% methanol), and polishing was carried out in the temperature range –30 to –25 °C, at a voltage of 20–25 V. Three foils were studied for each state. The average size of the structural elements was determined from the dark-field images. 

X-ray analysis was carried out on an Ultima IV diffractometer (Rigaku, Tokyo, Japan) by CuKα irradiation (40 kV and 30 mA; the slit size was 2 × 10 mm). The wavelength λ_Kα1_ = 1.54060 Å was used for calculations. The general view of the X-ray patterns was taken with a scanning step of 0.02° and exposure time at each point equal to 3 s. The phase composition of the alloy was determined via the Rietveld refinement method using MAUD (Material Analysis Using Diffraction) 1.993 for Windows [36]. Rietveld reference-free full-profile analysis was conducted with a procedure for minimizing the deviation between the experimental and calculated X-ray. The dislocation density was calculated by processing the data of X-ray diffraction analysis in MatLab R2022b.

Tensile testing was carried out at T = 20 °C on an Instron 5982 machine (Instron Engineering Corporation, Buckinghamshire, UK). The strain rate was 10^−3^ s^−1^. The force measurement accuracy was 1%. Samples with a gauge diameter of 3 mm were used. Three samples were tested for each state. 

Tensile-compressive fatigue tests were carried out on an Instron 8801 machine (Instron Engineering Corporation, Buckinghamshire, UK) at room temperature under conditions of a symmetrical loading cycle R = −1 with a frequency f = 30 Hz and a baseline of 10^7^ cycles. We used smooth samples of circular cross-sections and a working part diameter of 3 mm. 

## 3. Results

### 3.1. Microstructure of the Alloy in the Initial State and after Annealing at 600°C for 1 h, WQ

The microstructure of the Ti-15Mo alloy in the initial state is shown in Figure 2a. The SEM images show particles of the primary α_p_-phase with a mean size of 1.2 ± 0.3 µm, the volume fraction of which is (23 ± 3.5)%. The primary α_p_-phase of a globular morphology formed as a result of a slow cooling of the alloy from a temperature above the β-transus during the fabrication of the rods [26]. Taking into account that the β-grain boundaries are the predominant sites for the precipitation of α-particles, the mean size of β-grains was estimated to be 2.0 ± 0.5 μm. In the longitudinal section, the orientation of the particles of the primary α_p_-phase in the direction of deformation was found. Such a metallographic texture is associated with the method of obtaining the initial ingots [26].

Annealing at 600 °C, followed by water quenching, led to partial recrystallization of β-phase grains and the precipitation of the secondary α_s_-phase of lamellar morphology in the bulk of β-grains (Figure 2b). The structure is inhomogeneous, the size distribution of β-grains is bimodal, small grains of the β-phase with a mean size of 2 ± 0.4 µm and large recrystallized β-grains with a mean size of 8 ± 0.6 µm are observed. In this case, there were practically no quantitative changes in the phase composition of the alloy, so the volume fraction of the α-phase was (20 ± 3)%, and it is likely that the increase in the volume fraction due to the precipitation of the secondary αs-phase is compensated by the partial dissolution of the primary α_p_. The mean particle size of the primary α_p_-phase was 1.5 ± 0.5 µm. This structure was the original before the SPD.

### 3.2. Microstructure of the Alloy Subjected to ECAP-C

Figure 3a shows a typical SEM image of the alloy microstructure after ECAP-C in the longitudinal section. The structure is heterogeneous. It can be seen that the deformation by ECAP-C led to the fragmentation of β-grains and the primary α_p_ phase compared to the state after annealing at 600 °C, 1 h, WQ (Figure 2c). The boundaries of β-grains are not amenable to clear etching. Both globular and elongated grains of the primary α_p_-phase oriented in the direction of deformation are observed. In the bulk of β-grains, particles of the secondary α_s_-phase of lamellar morphology are visible. Figure 3b–d show TEM images of the fine structure of the alloy in the longitudinal section of the ingot after ECAP-C, which are characterized by a complex inhomogeneous contrast, indicating high internal stresses. The structure is saturated with dislocations; complex dislocation configurations in the form of dislocation cells are observed (Figure 3b). Figure 3c,d show images of particles of the primary and secondary α-phase after deformation. In the grains of the primary α_p_-phase, the average transverse size of which, determined from dark-field TEM images (Figure 3c), was 450 ± 30 nm, clusters of dislocations and dislocation cells are observed. Figure 3d shows an image of plates of the secondary α_s_ phase that are highly fragmented as a result of deformation. The average width of the α_s_ plates was 60 ± 10 nm, and the length was 600 ± 40 nm. At the same time, clusters of dislocations and dislocation walls are observed inside the plates, dividing the plate into fragments, the size of which is about 100 nm.

Figure 4a shows the initial β-grain separated into fragments separated by thin boundaries. Their mean size was 200 ± 30 nm. Figure 4b shows refined β-grains with a mean size of 150 ± 30 nm. In addition, equiaxed grains of the β-phase were found in the microstructure, probably formed as a result of dynamic recrystallization, then subjected to deformation (Figure 4c,d). Their mean size, determined from dark-field images, was 200 ± 20 nm. A developed substructure in the form of dislocation walls was observed in the bulk of the grains.

Figure 5 shows the XRD patterns of the Ti-15Mo alloy after annealing at 600 °C for 1 h and after ECAP-C. Peak broadening is observed, indicating an increase in internal stresses. In addition, redistribution of the peak intensities between the β- and α-phases takes place. With the help of X-ray phase analysis, it was found that during ECAP-C, additional precipitation of the α-phase occurs, and its volume fraction increased to 30 ± 3.5%. According to X-ray diffraction data, the density of dislocations in β-grains and α-grains was (6.3 and 3.4) × 10^15^ m^−2^, respectively.

### 3.3. Microstructure of an Alloy Subjected to ECAP-C and Subsequent Drawing

Figure 6 shows a typical microstructure of the Ti-15Mo alloy subjected to ECAP-C and drawing. It can be seen that the deformation by drawing led to additional refinement of the structure compared to ECAP-C. The orientation of structural elements along the axis of the rod is observed.

Figure 6b shows elongated particles of the primary α_p_-phase, located in the β-matrix, also subjected to fragmentation, and their transverse size is 180 ± 20 nm. Figure 6c shows elongated grains/subgrains of the β-phase, which alternate with interlayers of the secondary α_s_-phase of lamellar morphology. Inside the bands of both the β- and α_s_-phases, a developed substructure is observed in the form of fragments and dislocation cells (Figure 6c,d). The average width of fragments of the β-phase is 90 ± 10 nm, and of the α_s_-phase, −30 ± 5 nm. The volume fraction of the α-phase was 35 ± 2.0%. The XRD pattern of the Ti-15Mo alloy after drawing is presented in Figure 7. The density of dislocations in β- and α-grains is (6.4 and 3.7) × 10^15^ m^−2^, respectively.

Subsequent stress-relief annealing did not lead to significant changes in the microstructure (Figure 8). The average width of fragments of the β-phase and α_s_-phase remained practically unchanged and amounted to 90 ± 5 nm and 35 ± 5 nm, respectively. The change in the phase composition after annealing occurs within the error of the X-ray phase analysis method (the volume fraction of the α-phase was 37 ± 3.2%). According to the results of the X-ray structural analysis (Figure 7), it was found that annealing led to a decrease in the dislocation density both in the grains of the β- and α-phases ((5.8 and 2.6) × 10^15^ m^−2^, respectively).

### 3.4. Mechanical Properties of the Alloy after ECAP-C and Drawing

Figure 9 shows the results of mechanical tensile testing of the Ti-15Mo alloy at each stage of combined thermomechanical processing. Heating at a temperature of 600 °C followed by water quenching led to a noticeable increase in uniform (from 5 ± 0.8 to 10 ± 0.5%) and fracture (from 17 ± 1 to 24 ± 0.8%) elongations. At the same time, the decrease in strength was insignificant (from 1017 ± 10 to 980 ± 15 MPa). This combination of strength and ductility provided a satisfactory deformability during ECAP-C.

According to the results of mechanical tensile tests, it was found that the strength level in the alloy subjected to ECAP- increased from 980 ± 15 MPa to 1270 ± 20 MPa due to the formation of a UFG structure with a high density of crystal defects (Figure 9). Probably, an increase in the length of interfacial boundaries due to an increase in the volume fraction of the α phase also contributes to an increase in the strength of the alloy [37,38]. During tensile testing of the sample at room temperature, a strong localization of deformation was observed, which was accompanied by a significant decrease in uniform elongation (less than 2%). At the same time, the fracture elongation of the sample was 12%, which is sufficient technological plasticity for the implementation of subsequent drawing in order to further strengthen the alloy.

After drawing, the strength of the alloy increased to 1580 ± 10 MPa, but the ductility decreased to 8.4 ± 0.5%. It can be seen in Figure 9b that SPD processing also resulted in an increase in the alloy’s yield strength σ_0.2_. Subsequent annealing to relieve internal stresses at 200 °C for 1 h led to an increase in both uniform and fracture elongations (σ_uts_ = 1590 ± 30 MPa, δ_un_ = 2.5 ± 0.5%, δ = 10 ± 0.5%). 

Figure 10 shows the results of tensile-compression fatigue tests of the alloy in the initial state subjected to ECAP-C and after the combined thermomechanical treatment. It can be seen that ECAP-C and subsequent drawing led to an increase in the resistance of both low-cycle and high-cycle fatigue. In the initial state, the fatigue limit was 500 ± 10 MPa, while after ECAP-C its value increased to 620 ± 10 MPa, and subsequent drawing and annealing led to an increase in the fatigue limit to 710 ± 10 MPa (Figure 10).

## 4. Discussion

In this work, for the first time, a combined thermomechanical treatment, including ECAP-C and drawing, was applied to the Ti-15Mo alloy to obtain long-length rods with a UFG structure with a complex of high mechanical properties. To avoid the precipitation of the ω-phase, which embrittles the alloy and critically increases the elastic modulus, the alloy was treated in a more thermodynamically stable two-phase (α + β) state [13,34].

It is shown that ECAP-C (four passes: two passes at 350 °C, two passes at 300 °C) led to a significant refinement of grains of the β-matrix. The size of β-grains decreased from 2 μm to 200 nm due to the fragmentation and the development of dynamic recrystallization during SPD (see Figure 4c,d). In addition, the β-phase underwent decomposition initiated by plastic deformation, as evidenced by the additional precipitation of thin (up to 60 nm wide) plates of the secondary α_s_-phase (Figure 3d). In this case, the average size of the weakly deformed primary α-phase decreased from 1.5 μm to 450 nm. Similar features of microstructure transformation during ECAP were observed in titanium alloys having a two-phase structure [39]. In particular, the accumulation of dislocations and fragmentation during ECAP-C processing is more intense in the more plastic and more deformable β-phase with a *bcc* lattice, compared to the *hcp* lattice of the α-phase, which has a limited number of dislocation glide systems [13,40]. The isolation of secondary particles of the α-phase as a result of the decomposition of the β-phase led to an increase in the total volume fraction from 20 to 30%, which, apparently, is associated with a decrease in the temperature of phase transformations due to the high density of defects in the crystal structure and high internal stresses, causing acceleration of diffusion processes of alloying elements [41]. In the initial state, the temperature of the β→α phase transformation of the Ti-15Mo alloy lies in the range of 450–500 °C [26].

The subsequent drawing and annealing led to the precipitation of dispersed particles (up to 35 nm) of the secondary α-phase due to the additional decomposition of the metastable β-phase, a decrease in the size of β-grains/subgrains to 90 nm, and of the primary α_p_-phase to 180 nm (see Figure 6 and Figure 7). In addition, drawing contributed to the formation of a developed metallographic texture, which was characterized by elongated elements along the material flow direction, which is typical for metals and alloys with this deformation scheme [42].

Table 2 presents data on the strength characteristics of the Ti-15Mo alloy after ECAP-C, drawing, and annealing, and after conventional treatments [29] with a final two-phase (α + β)-structure. Traditionally, the thermomechanical treatment of metastable β-titanium alloys includes rolling at temperatures above or below the β-transus, a series of recrystallization annealings, and aging in the (α + β)-region [28,29]. It can be seen that the UFG Ti-15Mo had a better combination of mechanical properties compared to processing by standard methods (Table 2). In particular, the strength (1590 MPa) of the UFG Ti-15Mo alloy was much higher than the alloy’s strength after conventional treatments (1280 and 1320 MPa) [29]. This high strength led to higher high-cycle fatigue values of the UFG alloy in comparison to the alloy after conventional treatments (710 and 670 MPa, respectively). On the one hand, the alloy after the combined treatment described in this paper had common structural features with the alloy after a conventional treatment [29]; it is a developed texture and a band duplex (α + β)-structure consisting of a deformed β matrix, partially recrystallized equiaxed β grains, and globular and elongated α particles. On the other hand, the average size of grains/subgrains of the β-phase and primary α_p_-phase in the alloy obtained by ECAP-C and drawing was noticeably smaller than during standard treatment (Table 2), which resulted in strength enhancement.

Thus, an increase in the strength of the alloy compared to the initial state (from 1017 to 1590 MPa) is caused by the mechanisms of dislocation and grain boundary strengthening [16,17], as well as phase hardening due to additional precipitation of secondary particles of the “solid” α-phase, which increases the length of interphase boundaries [37]. However, at a satisfactory level of elongation to failure (10%), the alloy had a low value of uniform elongation (2.5%). Such mechanical behavior is characteristic of many metals and alloys after severe plastic deformation, when shear bands are nucleated in the UFG material [43,44].

An increase in the ultimate strength of the Ti-15Mo alloy due to the formation of a UFG two-phase structure provided an increase in the high-cycle fatigue limit from 500 in the initial state to 710 MPa (see Table 2). This effect is natural for many metals subjected to severe plastic deformation due to a decrease in grain size and an increase in tensile strength [19]. In UFG materials, the generation and movement of new dislocations are made difficult by the fine grain size and the presence of elastic stress fields caused by the high density of crystal defects introduced by SPD [19]. For this reason, the tension flow required for the development of microplastic deformation of the near-surface layer, which occurs at the stage of microfluidity in the initial period of cyclic testing, and the fatigue limit increase with decreasing grain size [19].

In this case, the increase in the endurance limit of the UFG Ti-15Mo alloy to 710 MPa is probably associated, on the one hand, with an increase in the resistance of the material to crack initiation, which is due to an increase in the strength of the material [18,19,20,21,22,23,24,25]. On the other hand, an increase in the length of the grain boundaries in the structure, which are an effective barrier and affect the tortuosity of the fatigue crack trajectory, restrains the propagation of the main crack [18]. An additional contribution to the nature and rate of crack propagation at low cyclic stresses is also made by such microstructural features as the type of dislocation glide systems, sizes of grains/subgrains, dislocation cells, and characteristics of particles of the second phase, including their average size, coherence, volume fraction, nucleation sites, and the distance between them and the crystallographic texture [18]. In this work, secondary α-particles additionally precipitated during ECAP-C and drawing of the Ti-15Mo alloy, which are difficult to deform compared to the β-matrix [13], probably become additional barriers to the propagation of the main crack, increasing the resistance of the material to its growth. An additional contribution to the increase in fatigue resistance can be made by the emerging crystallographic texture, but this requires a separate study.

Thus, the formation of a duplex UFG structure in the Ti-15Mo alloy with an average grain/subgrain size of the α_s_-phase and β-phase less than 100 nm by a combined deformation treatment, including ECAP-C and drawing, made it possible to increase the endurance limit by 42% and reach record values of 710 MPa. Such values are comparable with the endurance limit values of the (α + β)-UFG Ti-6Al-4V ELI alloy widely used in medicine (690 MPa) [21]. However, the Ti-15Mo alloy, in contrast to Ti-6Al-4V, does not contain elements toxic to the human body, so it may be preferable for implants that have been in the human body for a long time [45]. Such a combination of high mechanical properties and exceptional biocompatibility makes UFG Ti-15Mo alloy a promising material for the manufacture of heavily loaded implant parts subjected to long-term cyclic loads.

## 5. Conclusions

1. It is shown that the combined thermomechanical treatment, including ECAP-C (4 passes: two passes at 350 °C, two passes at 300 °C) and drawing (compression ratio 60%, at 300 °C) led to the formation of duplex UFG structure structures with an average grain/subgrain size of the β-phase of 90 nm in the rods of the Ti-15Mo alloy. The volume fraction of the α-phase was 37%, and the sizes of the primary α_p_- and secondary α_s_-phases were 180 nm and 35 nm, respectively.

2. It has been established that the formation of a two-phase UFG structure in the Ti-15Mo alloy leads to an increase in the ultimate strength from 1017 to 1590 MPa while maintaining ductility at the level of 10%.

3. It is shown that the achievement of high strength and ductility in long rods made of UFG Ti-15Mo alloy provides a record increase in the endurance limit (σ_−1_ = 710 MPa) under tension-compression conditions.

## Figures and Tables

**Figure 1 materials-15-08666-f001:**
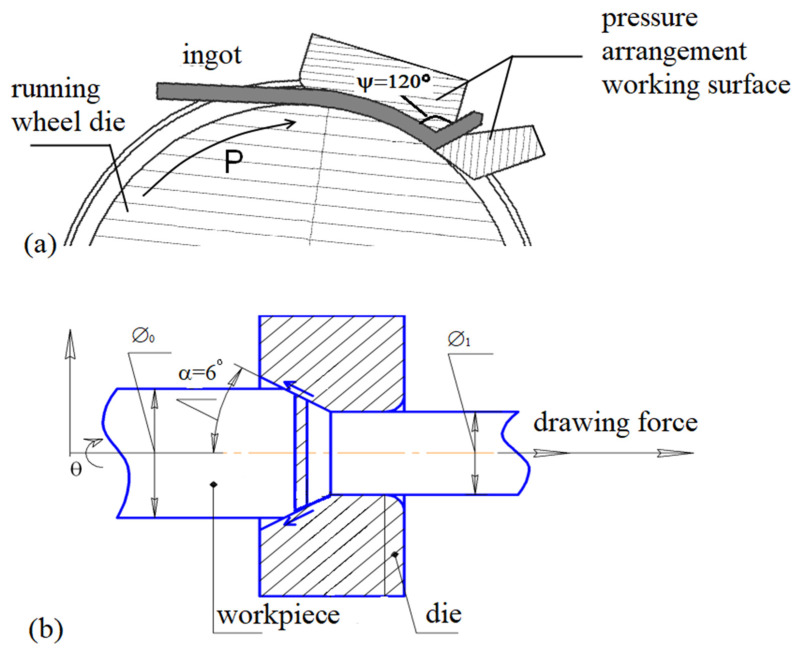
Schematic diagram of the combined processing: (**a**) ECAP-Conform process; (**b**) drawing process.

**Figure 2 materials-15-08666-f002:**
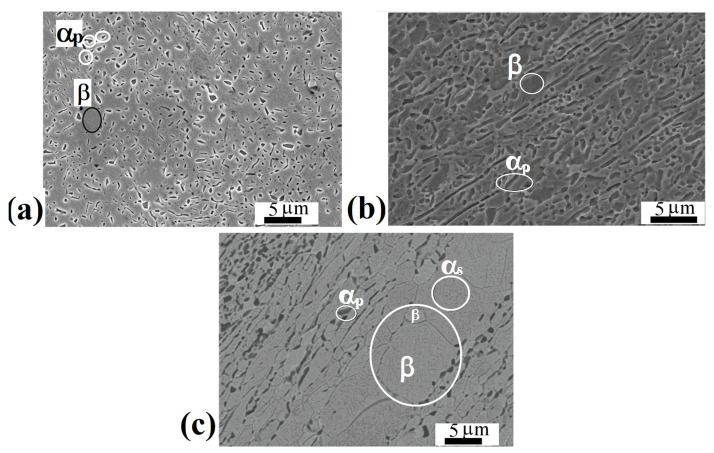
SEM images of the Ti-15Mo alloy: (**a**) in the initial state, the cross-section of the rod; (**b**) in the initial state, the longitudinal section of the rod; (**c**) after annealing at 600 °C, 1 h, WQ, longitudinal section of the rod.

**Figure 3 materials-15-08666-f003:**
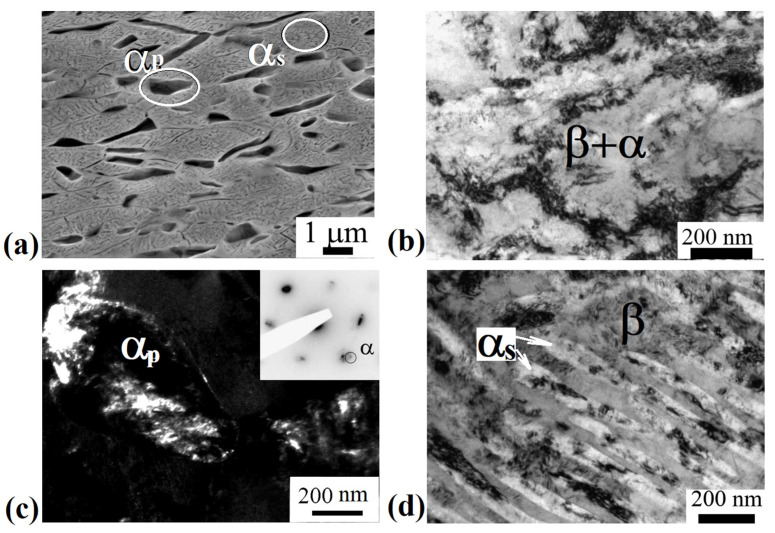
Microstructure of the Ti-15Mo alloy after ECAP-C, longitudinal section: (**a**) SEM image; (**b**) dislocation cells, TEM; (**c**) dark-field image of the primary α_p_ phase, TEM; (**d**) bright-field image of secondary α_s_-phase particles, TEM.

**Figure 4 materials-15-08666-f004:**
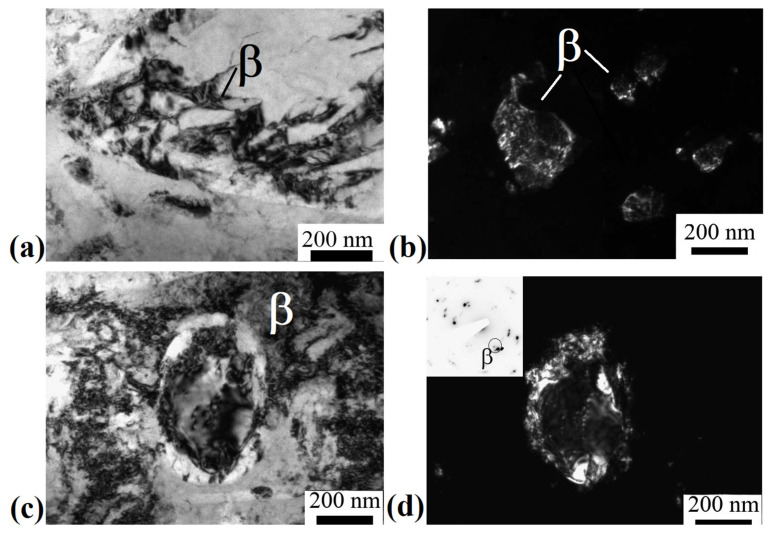
Microstructure of the alloy after ECAP-C, longitudinal section, TEM: (**a**) fragmented β-grain; (**b**) dark-field image of β-grains subjected to deformation; (**c**) bright field image of a recrystallized β grain; (**d**) dark field image of a recrystallized β grain.

**Figure 5 materials-15-08666-f005:**
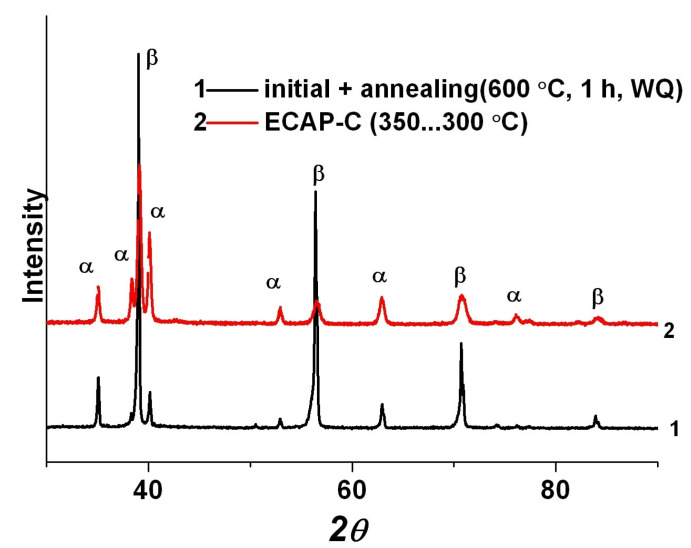
XRD patterns of the Ti-15Mo alloy after annealing at 600 °C for 1 h and after ECAP-C.

**Figure 6 materials-15-08666-f006:**
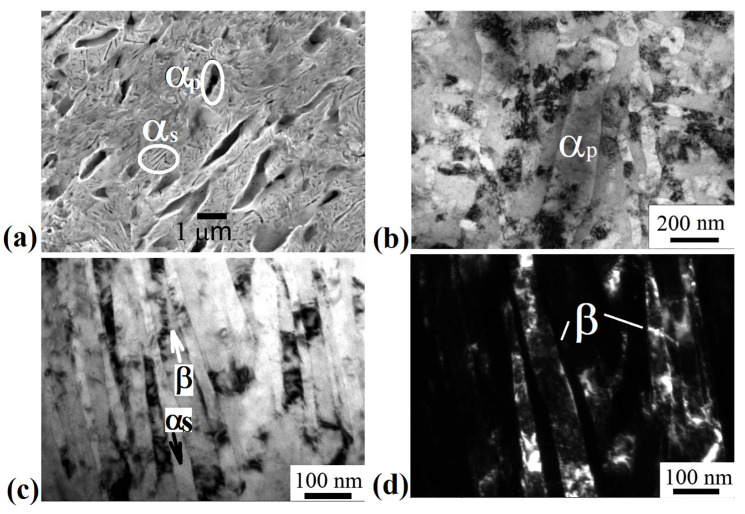
Microstructure of the Ti-15Mo alloy after ECAP-C and drawing: (**a**) SEM image, longitudinal section; (**b**) bright-field image of a grain of the primary α_p_ phase, cross-section; (**c**) bright-field image of β-grains and grains of the secondary α_s_-phase, longitudinal section; (**d**) dark-field image of β-phase grains, longitudinal section.

**Figure 7 materials-15-08666-f007:**
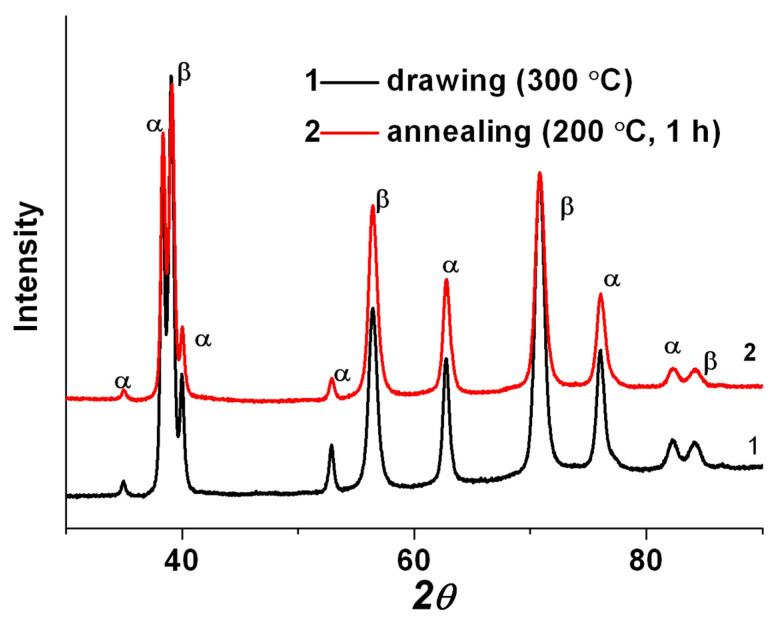
XRD patterns of the Ti-15Mo alloy after drawing and stress-relief annealing at 200 °C for 1 h.

**Figure 8 materials-15-08666-f008:**
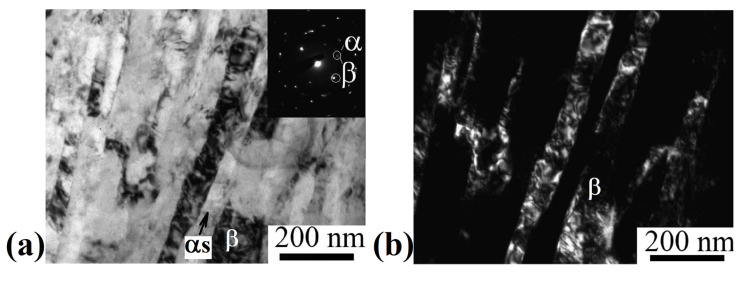
Microstructure of the Ti-15Mo alloy after ECAP-C, drawing and annealing to relieve internal stresses, longitudinal section: (**a**) bright-field image; (**b**) dark-field image of β-phase grains. TEM.

**Figure 9 materials-15-08666-f009:**
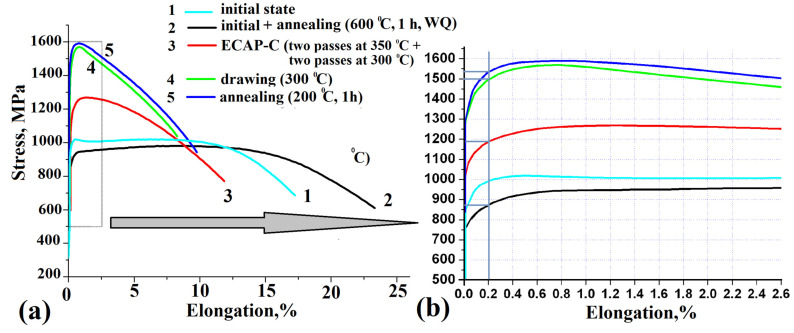
Results of the tensile mechanical tests for the Ti-15Mo alloy at each stage of the combined thermomechanical treatment: (**a**) typical engineering stress−strain curves; (**b**) an enlarged fragment of engineering stress−strain curves to an elongation of 2.6%.

**Figure 10 materials-15-08666-f010:**
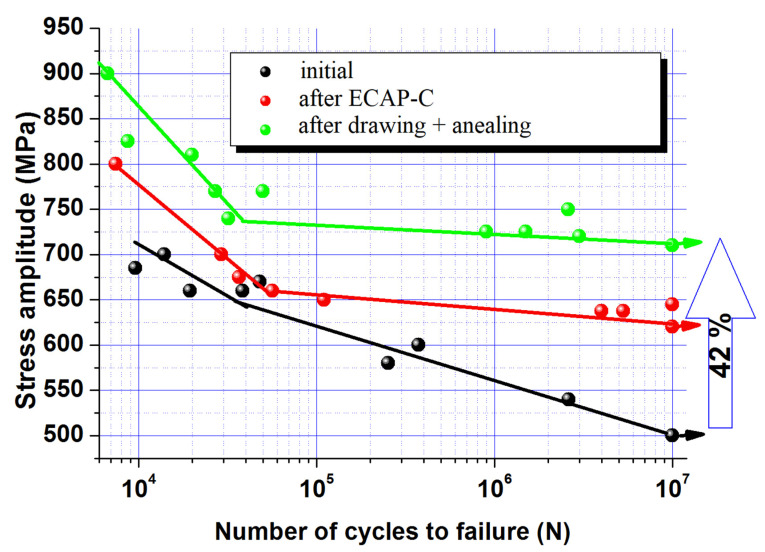
Fatigue curves of the Ti-15Mo alloy in the initial state, after ECAP-C, and after the combined thermomechanical treatment, including ECAP-C and drawing.

**Table 1 materials-15-08666-t001:** Chemical composition of the studied alloy (wt%).

Ti	Mo	O	Fe	C	N
balance	15.2	0.16	0.02	0.008	0.10

**Table 2 materials-15-08666-t002:** Microstructure parameters and mechanical properties of the Ti-15Mo alloy after combined deformation treatment, including ECAP-C and drawing and conventional treatments [29].

Treatment	D_β_, µm	D_α_, µm	σ_uts_, MPa	δ, %	σ_−1_, MPa, N = 10^7^
α + β, Initial state	2	1.2	1017	24	500
β, Continuous rolling Mill + (α + β) annealing + aging 480 °C, 4 h [29]	2	1	1280	14	670
α + β, hand rolling Mill + (α + β) annealing + aging 480 °C, 4 h [29]	2	1	1320	9	670
α + β, ECAP (two passes at 350 °C and two passes at 300 °C) + drawing 300 °C + 200 °C, 1 h	0.09	0.18	1590	10	710

## Data Availability

Not applicable.

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
