# Peer review of "Microstructure and Mechanical Properties of β-Titanium Ti-15Mo Alloy Produced by Combined Processing including ECAP-Conform and Drawing"

_materials, 2022, doi:10.3390/ma15238666_

Round 1

Reviewer 1 Report

The authors investigated the effect of heat treatment on the metallographic structure and mechanical properties of (α+β)-titanium alloys. And the possibility of processing large rods with ultrafine-grained structures was demonstrated. As a material with high corrosion resistance and superelasticity, titanium alloys have high application value in the medical and aerospace fields. The authors’ research is valuable for the metal treatment and production of the Ti15Mo in semi-finished products. The principle of equa-channel angular pressing (ECAP) has been resolved from the phase transformation perspective. Here are some of the problems with this article.

1.     Page 1: The state of the technology present can be expanded with at least 5 years of references (from 2017 to present). The recent studies on metastable β-titanium alloys should be appropriately supplemented.

2.     Page 3, in line 96: The author describes the deformation of the ingot according to route Bc, but I do not see it marked in Figure 1.

3.     Page 3, in lines 106-108: this paragraph makes little sense and should not exist in the article, the author should check this part again.

4.     Page 4, in line 189: I did not find where Figure 4e is in the article, which should be a description error. In addition, the text and labels in Figure 4 should be in a different color than the image’s background color, making it difficult for me to identify the metallographic structure described by the author.

5.     Similarly, the label colors should be reelected in Figures 2, 6 and 7.

6.     Page 8, in lines 263-267: Although the authors claim that the deformation appears strongly localized, in my opinion, this is caused by the transition from elastic to plastic deformation. Whereas ECAP-C is a new process for making ultrafine-grained materials by plastic deformation, the author's description of this phenomenon is inappropriate. A distinction should be made between the two stages of metallurgical changes, which should not be considered the same.

7.     In addition, I suggest that the authors should insert a partially enlarged drawing in Figure 8 that should include the stress-strain data up to 2.5% of the elongation. In this way, I can observe the yield point variation trend.

8.     Page 9, In Discussion: The authors compared the properties of Ti15Mo obtained after ECAP and drawing treatment with those obtained after conventional heat treatment. However, the effect of specific heat treatment parameters on the metallurgical transformation is not summarized, and only the drawing time is specified according to the existing standards. This is where I think the current research is incomplete.

Reviewer 2 Report

Editor, materials

Title: “Microstructure and Mechanical Properties of β-titanium Ti-2 15Mo Alloy Produced by Combined Processing Including 3 ECAP-Conform and Drawing”

.

Manuscript Number: materials -2057134

Dear Editor,

        I am attaching my review comments of the manuscript on a paper entitled “Microstructure and Mechanical Properties of β-titanium Ti-2 15Mo Alloy Produced by Combined Processing Including 3 ECAP-Conform and Drawing”.

.

In this paper, the authors have studied the influence of severe plastic deformation SPD followed by drawing on the microstructure evolution and the mechanical properties of β-titanium Ti-2 15Mo Alloy. The paper indicates the grain refinement and phase fragmentation of the material with the increase of the imposed strain that increases the tensile strength. Moreover, the annealing at 200 °C for one h improves the ductility and strength without any change or growth in the grain and phase sizes. Moreover, the endurance limit improved by 42%. The results of the paper were well presented and discussed. It is exciting research; the reviewer suggests accepting this paper for publication in the materials after a minor revision to cover the following comments. 

1-      Please rewrite the abstract it must be comprehensive and contain some details about aims, experimental work, results, and conclusion.

2-      Figure 1 must be enhanced, and the ECAP angles added. Moreover, the engineering drawing of the drawing die must be added to the figure.

3-      The total imposed strain value on the final bar must be added to the paper.

4-      Please clarify if the tensile curves are engineering stress-strain or true stress-strain. 

5-      The results of the initial case or even after annealing must be numbered as 3.1, then other sections continue in numbering.

6-      In line 175, please write elongations. Furthermore, why use relative elongation? It is better to use fracture elongation.

7-      αp-phase and αs phase compositions must be mentioned.

8-      The authors speak about XRD in material and methods on page 4, line 133, and results, page 7, line 217. Please insert the XRD patterns.

9-      It would be better to merge figures 1 and 8 into one figure. See, even the format of the two curves is different (please see the titles of the two curves).

10-  The title of table 2 must be modified as it contains previous results, not only the present one.

11-  Can the authors explain how the percentage of the phase is calculated, as mentioned in conclusion 2?                                                                   

                                                              Sincerely yours,
